# MODE NORMALIZATION

**Lucas Deecke, Iain Murray & Hakan Bilen**
University of Edinburgh
{l.deecke,i.murray,h.bilen}@ed.ac.uk

## ABSTRACT

Normalization methods are a central building block in the deep learning toolbox. They accelerate and stabilize training, while decreasing the dependence on manually tuned learning rate schedules. When learning from multi-modal distributions, the effectiveness of batch normalization (BN), arguably the most prominent normalization method, is reduced. As a remedy, we propose a more flexible approach: by extending the normalization to more than a single mean and variance, we detect modes of data on-the-fly, jointly normalizing samples that share common features. We demonstrate that our method outperforms BN and other widely used normalization techniques in several experiments, including single and multi-task datasets.

## 1 INTRODUCTION

A challenge in optimizing deep learning models is the change in input distributions at each layer, complicating the training process. Normalization methods, such as batch normalization (BN, Ioffe & Szegedy, 2015) aim to overcome this issue — often referred to as internal covariate shift (Shimodaira, 2000).[1] When applied successfully in practice, BN enables the training of very deep networks, shortens training times by supporting larger learning rates, and reduces sensitivity to parameter initializations. As a result, BN has become an integral element of many state-of-the-art machine learning techniques (He et al., 2016; Silver et al., 2017).

It can be difficult to standardize the activations in a neural network exposed to heterogeneous or multi-modal data. When training a deep neural network on images that come from a diverse set of visual domains, each with significantly different statistics, BN is not effective at normalizing the activations with a single mean and variance (Bilen & Vedaldi, 2017). In this paper we relax the assumption that the entire mini-batch should be normalized with the same mean and variance.

Our new normalization method, mode normalization (MN), first assigns samples in a mini-batch to different modes via a gating network, and then normalizes each sample with estimators for its corresponding mode (Figure 1). We further show that MN can be incorporated into other normalization techniques such as group normalization (GN, Wu & He, 2018) by learning which filters should be grouped together. The proposed methods can easily be implemented as layers in standard deep learning libraries, and their parameters are learned jointly with the other parameters of the network in an end-to-end manner. We evaluate MN on multiple classification tasks where it achieved a consistent improvement over currently available normalization approaches.

## 2 RELATED WORK

**Normalization.** Normalizing input data (LeCun et al., 1998) or initial weights of neural networks (Glorot & Bengio, 2010) are known techniques to support faster model convergence, and have been studied extensively. More recently, normalization has become part of functional layers that adjust the internal activations of neural networks. Local response normalization (LRN) (Lyu & Simoncelli, 2008; Jarrett et al., 2009) is used in various models (Krizhevsky et al., 2012; Sermanet et al., 2014) to perform normalization in a local neighborhood, and thereby enforce competition

---

[1]The underlying mechanisms are still being explored from a theoretical perspective (Bjorck et al., 2018; Kohler et al., 2018; Santurkar et al., 2018).

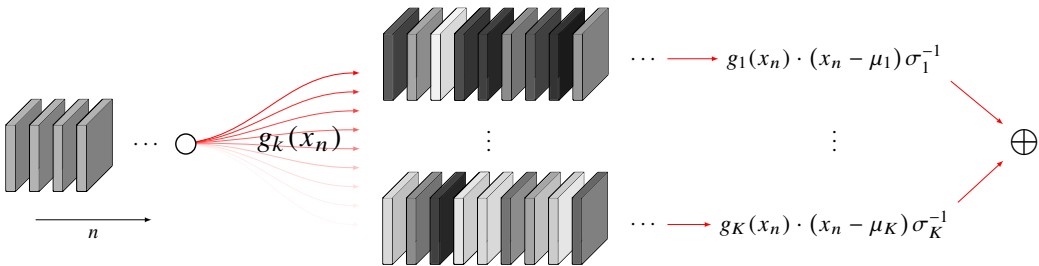

Figure 1: In mode normalization, incoming samples $\{x_n\}_{n=1}^N$ are weighted by a set of gating functions $\{g_k\}_{k=1}^K$. Gated samples contribute to component-wise estimators $\mu_k$ and $\sigma_k$, under which the data is normalized. After a weighted sum, the batch is passed on to the next layer. After training, the shifts and scales $\{\mu_k, \sigma_k\}_{k=1}^K$ are taken from running averages rather than batch statistics.

between adjacent pixels in a feature map. BN (Ioffe & Szegedy, 2015) implements a more global normalization along the batch dimension. Unlike LRN, BN requires two distinct training and inference modes of operation. At training time, samples in each batch are normalized with the batch statistics, while during inference samples are normalized using precomputed statistics from the training set. Heterogeneity and small batch sizes can both lead to inconsistencies between training and test data. Our proposed method alleviates the former issue by better dealing with different modes in the data, simultaneously discovering these and normalizing the data accordingly.

Several recent normalization methods (Ba et al., 2016; Ulyanov et al., 2017; Ioffe, 2017) have emerged that perform normalization along the channel dimension (Ba et al., 2016), or over a single sample (Ulyanov et al., 2017) to overcome the limitations of BN. Ioffe (2017) proposes a batch renormalization strategy that clips gradients for estimators by using a predefined range to prevent degenerate cases. While these methods are effective for training sequential and generative models respectively, they have not been able to reach the same level of performance as BN in supervised classification. Alongside these developments, BN has started to attract attention from theoretical viewpoints (Kohler et al., 2018; Santurkar et al., 2018).

More recently, Wu and He (Wu & He, 2018) have proposed a simple yet effective alternative to BN by first dividing the channels into groups and then performing normalization within each group. The authors show that group normalization (GN) can be coupled with small batch sizes without any significant performance loss, and delivers comparable results to BN when the batch size is large. We build on this method in Section 3.2, and show that it is possible to automatically infer filter groupings.

An alternative normalization strategy is to design a data-independent reparametrization of the weights in a neural network by implicitly whitening the representation obtained at each layer (Desjardins et al., 2015; Arpit et al., 2016). Lastly, Kalayeh & Shah (2019) propose fitting a Gaussian mixture model to account for the modality in intermediate feature distributions of some layers. While these methods show promising results, they do not generalize to arbitrary non-linearities and layers, or suffer from runtime restrictions.

**Mixtures of experts.** Mixtures of experts (MoE) (Jacobs et al., 1991; Jordan & Jacobs, 1994) are a family of models that involve combining a collection of simple learners to split up the learning problem. Samples are thereby allocated to differing subregions of the model that are best suited to deal with a given example. There is a vast body of literature describing how to incorporate MoE with different types of expert architectures such as SVMs (Collobert et al., 2002), Gaussian processes (Tresp, 2001), or deep neural networks (Eigen et al., 2013; Shazeer et al., 2017). Most similar to ours, Eigen et al. (2013) propose using a different gating network at each layer in a multilayer network to enable an exponential number of combinations of expert opinions. While our method also uses a gating function at every layer to assign the samples in a mini-batch to separate modes, it differs from the above MoE approaches in two key aspects: i) we use the assignments from the gating functions to normalize the data within a corresponding mode; ii) the normalized data is forwarded to a common module (i.e. a convolutional layer) rather than to multiple separate experts.

Our method is also loosely related to Squeeze-and-Excitation Networks (Hu et al., 2018), that adaptively recalibrate channel-wise feature responses with a gating function. Unlike their approach, we use the outputs of the gating function to normalize the responses within each mode.

**Multi-domain learning.** Our approach also relates to methods that parametrize neural networks with domain-agnostic and specific layers, and transfer the agnostic parameters to the analysis of very different types of images (Bilen & Vedaldi, 2017; Rebuffi et al., 2017; 2018). In contrast to these methods, which require the supervision of domain knowledge to train domain-agnostic parameters, our method can automatically learn to discover modes both in single and multi-domain settings, without any supervision. Recent work studies the role of adaptive intermediate features based on task-conditioned inputs (Perez et al., 2017). From this viewpoint MN can be understood as a conditional layer, however it has the different focus of accounting for modality in intermediate feature distributions.

## 3 METHOD

We first review the formulations of BN and GN in Section 3.1, and introduce our method in Section 3.2.

### 3.1 BATCH AND GROUP NORMALIZATION

Our goal is to learn a prediction rule $f \colon \mathcal{X} \to \mathcal{Y}$ that infers a class label $y \in \mathcal{Y}$ for a previously unseen sample $x \in \mathcal{X}$. For this purpose, we optimize the parameters of $f$ on a training set $\{x_i\}_{i=1}^{N_d}$ for which the corresponding label information $\{y_i\}_{i=1}^{N_d}$ is available, where $N_d$ denotes the number of samples in the data.

Without loss of generality, in this paper we consider image data as the input, and deep convolutional neural networks as our model. In a slight abuse of notation, we also use the symbol $x$ to represent the features computed by layers within the deep network, producing a three-dimensional tensor $\mathcal{X} = \mathcal{C} \times \mathcal{H} \times \mathcal{W}$ where the dimensions indicate the number of feature channels, height and width respectively. Batch normalization (BN) computes estimators for the mini-batch $\{x_n\}_{n=1}^{N}$ (usually $N \ll N_d$) by average pooling over all but the channel dimensions.[2] Then BN normalizes the samples in the batch as

$$\mathrm{BN}(x_n) = \alpha\Big(\frac{x_n - \mu}{\sigma}\Big) + \beta, \tag{1}$$

where $\mu$ and $\sigma$ are the mean and standard deviation of the mini-batch, respectively. The parameters $\alpha$ and $\beta$ are $|\mathcal{C}|$-dimensional vectors representing a learned affine transformation along the channel dimensions, purposed to retain each layer's representative capacity (Ioffe & Szegedy, 2015). This normalizing transformation ensures that the mini-batch has zero mean and unit variance when viewed along the channel dimensions.

Group normalization (GN) performs a similar transformation to that in (1), but normalizes along different dimensions. GN first separates channels $c = 1, \ldots, |\mathcal{C}|$ into fixed groups $G_j$, over which it then jointly computes estimators, e.g. for the mean $\mu_j = |G_j|^{-1} \sum_{x_c \in G_j} x_c$. Because GN does not average the statistics along the mini-batch dimension, it is appealing when it is not practical to use large batch sizes.

A potential problem when using GN is that channels that are being grouped together might get prevented from developing distinct characteristics in feature space. In addition, computing estimators from manually engineered rules as those found in BN and GN might be too restrictive under some circumstances, for example when jointly learning on multiple domains.

### 3.2 MODE NORMALIZATION

The heterogeneous nature of complex datasets motivates us to propose a more flexible treatment of normalization. Before the actual normalization is carried out, the data is first organized into modes to which it likely belongs. To achieve this, we reformulate the normalization in the framework of

---

[2]How estimators are computed is what differentiates many of the normalization techniques currently available. Wu & He (2018) provide a detailed introduction.

---

**Algorithm 1** Mode normalization, training phase.

---

**Input:** parameters $\lambda, K$, batch of samples $\{x_n\}$, small $\varepsilon$, learnable $\alpha, \beta, \Psi : \mathcal{C} \to \mathbb{R}^K$.

Compute expert assignments:

$$g_{nk} \leftarrow \big[\text{softmax} \circ \Psi(x_n)\big]_k$$

**for** $k = 1$ to $K$ **do**

Determine new component-wise statistics:

$$N_k \leftarrow \sum_n g_{nk}$$
$$\langle x \rangle_k \leftarrow \frac{1}{N_k} \sum_n g_{nk} x_n$$
$$\langle x^2 \rangle_k \leftarrow \frac{1}{N_k} \sum_n g_{nk} x_n^2$$

Update running means:

$$\overline{\langle x \rangle}_k \leftarrow \lambda \langle x \rangle_k + (1 - \lambda)\overline{\langle x \rangle}_k$$
$$\overline{\langle x^2 \rangle}_k \leftarrow \lambda \langle x^2 \rangle_k + (1 - \lambda)\overline{\langle x^2 \rangle}_k$$

**end for**
**for** $n = 1$ to $N$ **do**

Normalize samples with component-wise estimators:

$$\mu_k \leftarrow \langle x \rangle_k$$
$$\sigma_k^2 \leftarrow \langle x^2 \rangle_k - \langle x \rangle_k^2$$
$$y_{nk} \leftarrow g_{nk} \frac{x_n - \mu_k}{\sqrt{\sigma_k^2 + \varepsilon}}$$

**end for**
**Return:** $\{\alpha \sum_k y_{nk} + \beta\}_{n=1,\ldots,N}$

---

**Algorithm 2** Mode normalization, test phase.

---

**Input:** refer to Alg. 1.

Compute expert assignments:

$$g_{nk} \leftarrow \big[\text{softmax} \circ \Psi(x_n)\big]_k$$

**for** $n = 1$ to $N$ **do**

Normalize samples with running average of component-wise estimators:

$$\mu_k \leftarrow \overline{\langle x \rangle}_k$$
$$\sigma_k^2 \leftarrow \overline{\langle x^2 \rangle}_k - \overline{\langle x \rangle}_k^2$$
$$y_{nk} \leftarrow g_{nk} \frac{x_n - \mu_k}{\sqrt{\sigma_k^2 + \varepsilon}}$$

**end for**
**Return:** $\{\alpha \sum_k y_{nk} + \beta\}_{n=1,\ldots,N}$

---

**Algorithm 3** Mode group normalization.

---

**Input:** parameter $K$, sample $x \in \mathcal{C}$, small $\varepsilon$, learnable $\alpha, \beta, \Psi : \mathbb{R} \to \mathbb{R}^K$.

Compute channel-wise gates:

$$g_{ck} \leftarrow \big[\text{softmax} \circ \Psi(x_c)\big]_k$$

**for** $k = 1$ to $K$ **do**

Update estimators and normalize:

$$\mu_k \leftarrow \langle x \rangle_k$$
$$\sigma_k^2 \leftarrow \langle x^2 \rangle_k - \langle x \rangle_k^2$$
$$y_k \leftarrow \frac{x - \mu_k}{\sqrt{\sigma_k^2 + \varepsilon}}$$

**end for**
**Return:** $\frac{\alpha}{K} \sum_k y_k + \beta$

---

mixtures of experts (MoE). In particular, we introduce a set of simple gating functions $\{g_k\}_{k=1}^K$ where $g_k : \mathcal{X} \to [0, 1]$ and $\sum_k g_k(x) = 1$. In mode normalization (MN, Alg. 1), each sample in the mini-batch is then normalized under voting from its gate assignment:

$$\text{MN}(x_n) \triangleq \alpha \bigg( \sum_{k=1}^K g_k(x_n) \frac{x_n - \mu_k}{\sigma_k} \bigg) + \beta, \tag{2}$$

where $\alpha$ and $\beta$ are a learned affine transformation, just as in standard BN.[3] The mean $\mu_k$ and variance $\sigma_k$ estimates are weighted averages under the gating network. For example, the $k$'th mean is estimated from the batch as

$$\mu_k = \langle x \rangle_k = \frac{1}{N_k} \sum_n g_k(x_n) \cdot x_n, \tag{3}$$

where $N_k = \sum_n g_k(x_n)$. In our experiments, we average over $\mathcal{W}$ and $\mathcal{H}$, and parametrize the gating functions via an affine transformation $\Psi : \mathcal{C} \to \mathbb{R}^K$ of the channels, which is jointly learned alongside the other parameters of the network. This is followed by a softmax activation, reminiscent of attention mechanisms (Denil et al., 2012; Vinyals et al., 2015).

Mode normalization generalizes BN, which can be recovered as a special case by setting $K = 1$, or if the gates collapse: $g_k(x_n) = \text{const.} \ \forall \, k, n$. To demonstrate how the extra flexibility helps, Fig. 2

---

[3]Learning individual $\{(\alpha_k, \beta_k)\}_{k=1}^K$ for each mode did not improve performance in preliminary experiments.

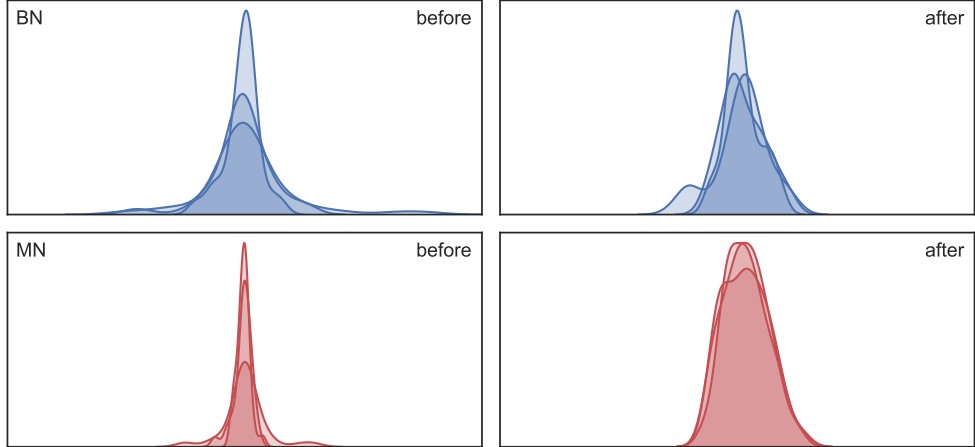

Figure 2: Channel-wise histograms for `conv3-64-1` in VGG13. Top row shows BN before and after the normalization is applied to samples from CIFAR10. The bottom row shows how the layer's distribution is transformed when MN is inserted into the network.

shows histograms over channels in layer `conv3-64-1` of VGG13 (Simonyan & Zisserman, 2015) for 1024 samples on CIFAR10, both before and after the normalization via BN and MN with $K = 2$. MN normalizes channels with multi-modal behavior, something that BN visibly struggles with.

As in BN, during training we normalize samples with estimators computed from the current batch. To normalize the data during inference (Alg. 2), we keep track of component-wise running estimates, borrowing from online EM approaches (Cappé & Moulines, 2009; Liang & Klein, 2009). Running estimates are updated in each iteration with a memory parameter $\lambda \in (0, 1]$, e.g. for the mean:

$$\overline{\langle x \rangle}_k = \lambda \langle x \rangle_k + (1 - \lambda)\overline{\langle x \rangle}_k. \tag{4}$$

Bengio et al. (2016) and Shazeer et al. (2017) propose the use of additional losses that either prevent all samples to focus on a single gate, encourage sparsity in the gate activations, or enforce variance over gate assignments. We did not find such penalties necessary for our MN experiments. Moreover, we wanted MN to be able to reduce to traditional BN, if that is the optimal thing to do. In practice, we seldom observed this behavior: gates tend to receive an even share of samples overall, and they are usually assigned to individual modes.

### 3.3 MODE GROUP NORMALIZATION

As discussed in Section 2, GN is less sensitive to the batch size (Wu & He, 2018). Here, we show that GN can also be adapted to normalize based on soft assignments into different modes. Unlike BN, GN computes averages over individual samples instead of the entire mini-batch. This makes slight modifications necessary, resulting in mode group normalization (MGN, Alg. 3). Instead of learning mappings with their pre-image in $\mathcal{X}$, in MGN we learn a gating network $g \colon \mathbb{R} \to \mathbb{R}^K$ that assigns channels to modes. After average-pooling over width and height, estimators are computed by averaging over channel values $x_c \in \mathbb{R}$, for example for the mean $\mu_k = \langle x \rangle_k = C_k^{-1} \sum_c g_k(x_c) \cdot x_c$, where $C_k = \sum_c g_k(x_c)$. Each sample is subsequently transformed via

$$\text{MGN}(x) \triangleq \frac{\alpha}{K} \sum_k \frac{x - \mu_k}{\sigma_k} + \beta, \tag{5}$$

where $\alpha$ and $\beta$ are learnable parameters for channel-wise affine transformations. One of the notable advantages of MGN (that it shares with GN) is that inputs are transformed in the same way during training and inference.

A potential risk for clustering approaches is that clusters or modes might collapse into one (e.g., Xu et al., 2005). Although it is possible to address this with a regularizer, it has not been an issue in

either MN or MGN experiments. This is likely a consequence of the large dimensionality of feature spaces that we study in this paper, as well as sufficient levels of variation in the data.

## 4 EXPERIMENTS

We consider two experimental settings to evaluate our methods: multi-task (Section 4.1) and single task (Section 4.2). All experiments use standard routines within PyTorch (Paszke et al., 2017).[4]

### 4.1 MULTI-TASK

**Data.** In the first experiment, we wish to enforce heterogeneity in the data distribution, i.e. explicitly design a distribution of the form $\mathbb{P} = \sum_d \pi_d \mathbb{P}_d$. We generated a dataset whose images come from diverse distributions by combining four image datasets: i) **MNIST** (LeCun, 1998) which contains grayscale scans of handwritten digits. The dataset has a total of 60 000 training samples, as well as 10 000 samples set aside for validation. ii) **CIFAR-10** (Krizhevsky, 2009) is a dataset of colored images that show real world objects of one of ten classes. It contains 50 000 training and 10 000 test images. iii) **SVHN** (Netzer et al., 2011) is a real-world dataset consisting of 73 257 training samples, and 26 032 samples for testing. Each image shows one of ten digits in natural scenes. iv) **Fashion-MNIST** (Xiao et al., 2017) consists of the same number of single-channel images as are contained in MNIST. The images contain fashion items such as sneakers, sandals, or dresses instead of digits as object classes. We assume that labels are mutually exclusive, and train a single network — LeNet (LeCun et al., 1989) with a 40-way classifier at the end — to jointly learn predictions on them.

**Mode normalization.** We trained for 3.5 million data touches (15 epochs), with learning rate reductions by 1/10 after 2.5 and 3 million data touches. We found that training for additional epochs did not notably improve performance. The batch size was $N = 128$, and running estimates were kept with $\lambda = 0.1$. We varied the number of modes in MN over $K = \{2, 4, 6\}$. Average performances over five random initializations as well as standard deviations are shown in Table 1. MN outperformed standard BN, as well as all other normalization methods.

The additional overhead of MN is small; even in our naive implementation, setting $K = 6$ resulted in roughly a 5% increase in runtime. However, increasing $K$ did not always improve the performance. Higher mode numbers are likely to suffer from estimating statistics from smaller and smaller partitions of the batch, a known issue in traditional BN. Experiments with larger batch sizes support this argument (Appendix A). In all remaining trials, which involve single datasets and deeper networks, we therefore fixed $K = 2$.

Table 1: Test set error rates (%) of batch norm (BN), instance norm (IN, Ulyanov et al., 2017), layer norm (LN, Ba et al., 2016), and mode norm (MN) in the multi-task setting for a batch size of $N = 128$. Shown are average top performances over five initializations alongside standard deviations. Additional results for $N = \{256, 512\}$ are shown in the Appendix.

| BN | IN | LN | MN | $K$ |
|---|---|---|---|---|
| $26.91 \pm 1.08$ | $28.87 \pm 2.28$ | $27.31 \pm 0.71$ | $\underline{23.16} \pm 1.23$ | 2 |
| | | | $24.25 \pm 0.71$ | 4 |
| | | | $25.12 \pm 1.48$ | 6 |

**Mode group normalization.** Group normalization is designed specifically for applications in which large batch sizes become prohibitive. We simulated this regime by reducing batch sizes to $N = \{4, 8, 16\}$, and trained each model for 50 000 gradient updates. We used the same configuration as before, except for a smaller initial learning rate $\gamma = 0.02$, which was reduced by 1/10 after 35 000 and 42 500 updates. In GN, we allocated two groups per layer, and accordingly set $K = 2$ in MGN. As a baseline, results for BN and MN were also included. Average performances over five initializations and their standard deviations are shown in Table 2. As previously reported by Wu & He (2018),

---

[4]Accompanying code is available under `github.com/ldeecke/mn-torch`.

BN failed to maintain its performance when the batch size is small during training. Though MN performed slightly better than BN, its performance also degraded in this regime. GN is more robust to small batch sizes, yet MGN further improved over GN, and — by combining the advantages of GN and MN — achieved the best performance for different batch sizes among all four methods.

Table 2: Test set error rates (%) for BN, MN, mode group norm (MGN) and group norm (GN) on small batch sizes. Shown are average top performances over five initializations alongside standard deviations.

| $N$ | BN | MN | GN | MGN |
|---|---|---|---|---|
| 4 | $33.40 \pm 0.75$ | $32.80 \pm 1.59$ | $32.15 \pm 1.10$ | $31.30 \pm 1.65$ |
| 8 | $31.98 \pm 1.53$ | $29.05 \pm 1.51$ | $28.60 \pm 1.45$ | $26.83 \pm 1.34$ |
| 16 | $30.38 \pm 0.60$ | $28.70 \pm 0.68$ | $27.63 \pm 0.45$ | $\underline{26.00} \pm 1.68$ |

## 4.2 SINGLE TASK

**Data.** Here mode normalization is evaluated in single-image classification tasks, showing that it can be used to improve performance in several recently proposed convolutional networks. For this, we incorporated MN into multiple modern architectures, and evaluated it on **CIFAR10** and **CIFAR100** datasets and then on a large-scale dataset, **ILSVRC12** (Deng et al., 2009). Unlike CIFAR10, CIFAR100 has 100 classes, but contains the same number of training images (600 images per class). ILSVRC12 contains around 1.2 million images from 1000 object categories.

**Network In Network.** Since the original Network In Network (NIN, Lin et al., 2014) does not contain any normalization layers, we modified the network architecture to add them, coupling each convolutional layer with a normalization layer (either BN or MN). We trained the resulting models on CIFAR10 and CIFAR100 for 100 epochs with SGD and momentum, using a batch size of $N = 128$. Initial learning rates were set to $\gamma = 10^{-1}$, which we reduced by 1/10 at epochs 65 and 80 for all methods. Running averages were stored with $\lambda = 0.1$. During training we randomly flipped images horizontally, and cropped each image after padding it with four pixels on each side. Dropout (Srivastava et al., 2014) is known to occasionally cause issues in combination with BN (Li et al., 2018), and reducing it to 0.25 (as opposed to 0.5 in the original publication) improved performance. For this large model, incorporating MN with $K = 2$ into NIN adds less than 1% to the number of trainable parameters.

We report the test error rates with NIN on CIFAR10 and CIFAR100 in Table 3 (left). NIN with BN obtains an error rate similar to that reported for the original network in Lin et al. (2014), while MN ($K = 2$) achieves an additional boost of 0.4% and 0.6% over BN on CIFAR10 and CIFAR100, respectively.

Table 3: Test set error rates (%) with BN and MN for NIN and VGG13.

| | Network In Network | | | VGG13 | |
|---|---|---|---|---|---|
| | Lin et al. | BN | MN | BN | MN |
| CIFAR10 | 8.81 | 8.82 | $\underline{8.42}$ | 8.28 | $\underline{7.79}$ |
| CIFAR100 | – | 32.30 | $\underline{31.66}$ | 31.15 | $\underline{30.06}$ |

**VGG Networks.** Another popular class of deep convolutional neural networks are VGG networks (Simonyan & Zisserman, 2015). In particular we trained a VGG13 with BN and MN on CIFAR10 and CIFAR100. For both datasets we optimized using SGD with momentum for 100 epochs, setting the initial learning rate to $\gamma = 0.1$, and reducing it at epochs 65, 80, and 90 by a factor of 1/10. The batch size was $N = 128$. As before, we set the number of modes in MN to $K = 2$, and keep estimators with $\lambda = 0.1$. When incorporated into the network, MN improves the performance of VGG13 by 0.4% on CIFAR10, and over 1% on CIFAR100, see Table 3 (right).

**Residual Networks.**    Unlike NIN and VGG, Residual Networks (He et al., 2016) originally included layer-wise batch normalizations. We trained a ResNet20 on CIFAR10 and CIFAR100 in its original architecture (i.e. with BN), as well as with MN ($K=2$), see Table 4 (left). On both datasets we followed the standard training procedure and trained both models for 160 epochs of SGD with momentum parameter of 0.9, and weight decay of $10^{-4}$. Running estimates were kept with $\lambda=0.1$, the batch size set to $N=128$. Our implementation of ResNet20 (BN in Table 4) performs slightly better than that reported in the original publication (8.42% versus 8.82%). Replacing BN with MN achieves a notable 0.45% and 0.7% performance gain over BN in CIFAR10 and CIFAR100, respectively.

Using the same setup as for ResNet20, we ran additional trials using a deeper ResNet56. As shown in Table 4 (right), replacing all normalization layers with MN resulted in an improvement over BN of roughly 0.5% on CIFAR10, and 1% on CIFAR100.

Table 4: Test error (%) for ResNet20, ResNet56 normalized with BN and MN.

|  | **ResNet20** | | | **ResNet56** | | |
| --- | --- | --- | --- | --- | --- | --- |
|  | **He et al.** | **BN** | **MN** | **He et al.** | **BN** | **MN** |
| CIFAR10 | 8.75 | 8.44 | 7.99 | 6.97 | 6.87 | 6.47 |
| CIFAR100 | – | 32.24 | 31.52 | – | 29.70 | 28.69 |

We also tested our method in the large-scale image recognition task of ILSVRC12. Concretely, we replaced BN in a ResNet18 with MN ($K=2$), and trained both resulting models on ILSVRC12 for 90 epochs. We set the initial learning rate to $\gamma=0.1$, reducing it at epochs 30 and 60 by a factor of 1/10. SGD was used as optimizer (with momentum parameter set to 0.9, weight decay of $10^{-4}$). To accelerate training we distributed the model over four GPUs, with an overall batch size of $N=256$. As can be seen from Table 5, MN results in a small but consistent improvement over BN in terms of top-1 and top-5 errors.

Table 5: Top-1 and top-5 error rates (%) of ResNet18 on ImageNet ILSVRC12, with BN and MN.

| **Top-$k$ Error** | **BN** | **MN** |
| --- | --- | --- |
| 1 | 30.25 | 30.07 |
| 5 | 10.90 | 10.65 |

**Qualitative analysis.**    In Fig. 3 we evaluated the experts $g_k(\{x_n\})$ for samples from the CIFAR10 test set in layers `conv3-64-1` and `conv-3-256-1` of VGG13, and show the samples that have been assigned the highest probability to belong to either of the $K=2$ modes. In the normalization belonging to `conv3-64-1`, MN is sensitive to a red-blue color mode, and separates images accordingly. In deeper layers, separations seem to occur on the semantic level. In `conv-3-256-1` for instance, MN separates smaller objects from those that occupy a large portion of the image.

## 5    CONCLUSION

Stabilizing the training process of deep neural networks is a challenging problem. Several normalization approaches that aim to tackle this issue have recently emerged, enabling training with higher learning rates, faster model convergence, and allowing for more complex network architectures.

Here, we showed that normalization approaches can be extended to allow the network to jointly normalize its features within multiple modes. We further demonstrated that accounting for modality in intermediate feature distributions results in a consistent improvement in classification performance for various deep learning architectures. As part of future work, we plan to explore customized, layer-wise mode numbers in MN, and automatically determining them, e.g. by using concepts from sparse regularization.

conv3-64-1

conv3-256-1

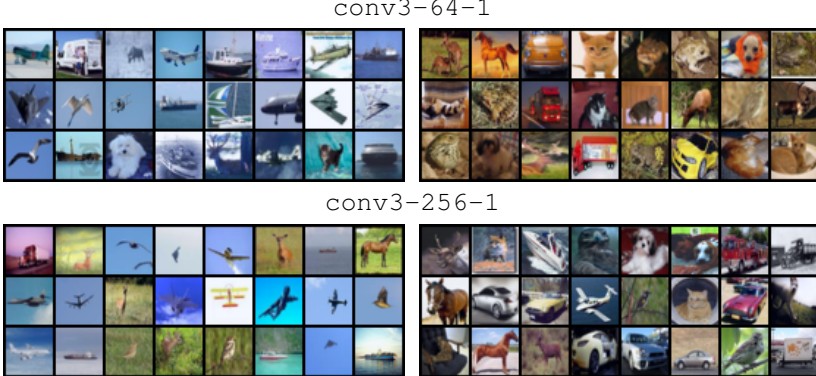

Figure 3: Test samples from CIFAR10 that were clustered together by two experts in an early layer (top) and a deeper layer (bottom) of VGG13.

## 6 ACKNOWLEDGMENTS

We gratefully acknowledge the support of Prof. Vittorio Ferrari and Timothy Hospedales for providing computational resources, and the NVIDIA Corporation for the donation of a Titan Xp GPU used in this research.

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

## A  ADDITIONAL MULTI-TASK RESULTS

Shown in Table 6 are additional results for jointly training on MNIST, CIFAR10, SVHN, and Fashion-MNIST. The same network is used as in previous multi-task experiments, for hyperparameters see Section 4. In these additional experiments, we varied the batch size to $N = \{256, 512\}$. For larger batch sizes, increasing $K$ to values larger than two increases performance, while for a smaller batch size of $N = 128$ (c.f. Table 1), errors incurred by finite estimation prevent this benefit from appearing.

Table 6: Test set error rates (%) of multiple normalization methods in the multi-task setting for large batch sizes. The table contains average performances over five initializations, alongside their standard deviation.

| $N$ | **BN** | **IN** | **LN** | **MN** | $K$ |
|-----|--------|--------|--------|--------|-----|
| 256 | $26.34 \pm 1.82$ | $31.15 \pm 3.46$ | $26.95 \pm 2.51$ | $25.29 \pm 1.31$ | 2 |
|     |        |        |        | $25.04 \pm 1.88$ | 4 |
|     |        |        |        | $\underline{24.88} \pm 1.24$ | 6 |
| 512 | $26.51 \pm 1.15$ | $29.00 \pm 1.85$ | $28.98 \pm 1.32$ | $26.18 \pm 1.86$ | 2 |
|     |        |        |        | $\underline{24.29} \pm 1.82$ | 4 |
|     |        |        |        | $25.33 \pm 1.33$ | 6 |

