# OpenReview forum: "Mode Normalization"
_ICLR.cc/2019/Conference_

### Official Review · AnonReviewer2 · 2018-11-02
**Might have already been published and pushes BN towards small mini-batches**

**Rating:** 6
**Confidence:** 4

**Review:**

Summary:
Batch Normalization (BN) suffers from 2 flaws: 1) It performs poorly when the batch size is small and 2) computing only one mean and one variance per feature might be a poor approximation for multi-modal features. To alleviate 2), this paper introduces Mode Normalization (MN) a new normalization technique based on BN. It uses a gating mechanism, similar to an attention mechanism, to project the examples in the mini-batch onto K different modes and then perform normalization on each of these modes.

Clarity:
The paper is clearly written, and the proposed normalization is well explained.

Novelty:
The proposed normalization is somewhat novel. I also found a similar paper on arXiv (submitted for review to IEEE Transactions on Pattern Analysis and Machine Intelligence, 2018): M. M. Kalayeh, M. Shah, Training Faster by Separating Modes of Variation in Batch-normalized Models, arXiv 2018. I didn’t took the time to read this paper in details, but the mixture normalization they propose seems quite close to MN. Could the authors comment on this?

Pros and Cons:
+ Clearly written and motivated
+ Try to address BN’s weakness, which is an important direction in deep learning
- I found similar papier in the literature
- The proposed method aims to make BN perform better, but pushes it toward small batch settings, which is where BN performs poorly.
- Misses comparisons with other techniques (see detailed comments).

Detailed Comments:
1. Multi-modality:
It is not clear if the features are multimodal when performing classification tasks. Some histograms of a few features in the network would have help motivate the proposed normalization. However, it seems indeed to be an issue when training GANs: to make BN work when placed in the discriminator, the real and fake examples must be normalized separately, otherwise the network doesn't train properly. Moreover, when dealing with multimodal datasets (such as the one you created by aggregating different datasets), one can use the FiLM framework (V. Dumoulin et al., Feature-wise transformations, Distill 2018), and compute different means and variances for each datasets. How would the proposed method perform against such method?
2. Larger scale:
It would be nice to see how MN performs on bigger networks (such as the ResNet50, or a DenseNet), and maybe a more interesting fully-connected benchmark, such as the deep autoencoder.
3. Small batch regime:
It seems that the proposed method essentially pushes BN towards a regime of smaller mini-batch size, where it is known to performs poorly. For instance, the gain in performances on the ImageNet experiments drops quite a lot already, since the training is divided on several GPUs (and thus the effective mini-batch is already reduced quite a lot). This effect gets worse as the size of the network increases, since the effective mini-batch size gets smaller. This problem also appears when working on big segmentation tasks or videos: the mini-batch size is typically very small for those problems. So I fear that MN will scale poorly on bigger setups. I also think that this is the reason why you need to use extremely small K.
4. Validation set:
What validation sets are you using in your experiments? In section 4.1, the different dataset and their train / test splits are presented, but what about validation?

Conclusion:
Given the similarity with another paper already in the literature, I reject the paper. Also, it seems to me that the technique actually pushed BN towards a small batch regime, where it is known to perform poorly. Finally, it misses comparison with other techniques.

Revision:
After the rebuttal, I increased my rating to a 6. I feel this paper could still be improved by better motivating why multi-modality is important for single tasks (for example, by plotting histograms of activations from the network). I also think that the paper by Kalayeh & Shah should be presented in more details in the related work, and also be compared to in the experimental setup (for example on a small network), especially because the authors say they have experience with GMMs.

---

> ### Author Response · Authors · 2018-11-15
> **Re: Might have already been published and pushes BN towards small mini-batches**
>
> Three main concerns were raised: (a.) a similar publication exists, giving grounds for a clear rejection of this paper. We thank the reviewer for bringing the interesting paper by Kalayeh & Shah to our attention, but show below that this claim is unjustified. (b.) MN suffers from weaknesses that BN also suffers from in the small batch size regime, and (c.) the paper should discuss some additional related methods.
>
> Regarding (a.): we are thankful for having this paper pointed out to us and will include it in our revision. That being said, we strongly rebut the claim that their paper is equivalent to ours, as their approach is very different. After reading their preprint in detail, we summarize below.
>
> The crucial difference is that in MN we employ a Mixture of Experts (MoE) approach and parametrize each expert with a simple attention-like mechanism on the image’s features. MN can effortlessly be added to any modern deep convolutional network, can be optimized with standard SGD, has a very small computational overhead, and introduces only a single hyperparameter (number of modes K). On the other hand, Kalayeh & Shah propose using a GMM to fit the feature distribution within the normalization unit (from hereon, we thus abbreviate MN-GMM). As it happens, we experimented with a GMM-based approach before designing MN, so we are well familiar with the several technical difficulties and impracticalities that using GMMs imposes:
>
> *  Due to the complexity of fitting GMMs, in their experiments Kalayeh & Shah never swap out all BN layers with MN-GMM layers, see p. 7 (right). So their resulting network is a mixture of BN and (very few, usually 1) MN-GMM normalizations. We designed MN to be lightweight and easy to deploy, and in our experiments show that MN can replace the entirety of BN layers, even in a deep network.
> * As Kalayeh & Shah explain on p. 6 (right column) they fit the GMM via EM, in a completely separate optimization step, outside the training loop of the network. In designing our method, it was important to us to sidestep this restriction, and MN can be trained end-to-end alongside the other parameters of the network.
> * Further complicating MN-GMM is that it requires careful, manual decisions in its tuning. From our own experiments, we are well aware of the considerations one needs to ponder over in MN-GMM. A few examples: (i.) how many EM iterations are needed? (ii.) Which BN units should be replaced, which should remain intact? (iii.) How should the GMM parameters be initialized? (iv.) How many components should be assumed? In MN, the practitioner needs to make a single choice (in that K needs to be set). Once that choice has been made, MN can be used off-the-shelf, making it straightforward to use in an applied setting.
>
> In MN-GNN Kalayeh & Shah (2018) propose an interesting modification to BN, however it should be clear from the above points that the similarities to our method are extremely limited. R2 states that “I didn’t took the time to read this paper in details”, only to continue “given the similarity with another paper already in the literature, I reject the paper”. We were very surprised by the rejection based on a “quick read”, and – for a top-tier conference like ICLR – would have found it appropriate to read the mentioned paper and to compare it to ours in a more careful manner. Once more, we firmly reject the implication that our proposed method has been covered in their publication, or that we, in any way, copied from their work.
>
> (b.): splitting up batches does introduce errors from finite estimation, which is an issue that we raise ourselves on p. 6, third paragraph. As we argue in our paper, many applications exist where the batch size restriction isn’t a major issue, and a larger error results from the underlying modality of the task. MN is aimed at alleviating issues in these particular tasks, we never designed it to solve the small batch size issues of BN, and at no point claim that it does.
>
> That being said, even though MN splits minibatches into multiple modes by construction (thereby collecting statistics from less samples than BN), in practice MN still performs better than BN, even for small batch sizes. This is shown in Table 2, where MN clearly is more robust to smaller batch sizes than BN.
>
> (c.): FiLM learns to adaptively influence the output of a neural network by applying transformations to intermediate features conditioned on some input. FiLM’ed networks still use BN, and thus FiLM does not address any shortcomings of BN, so MN can simply be used alongside FiLM. There is a weak connection to our paper in that MN can also be seen as a conditional layer, however with the completely different focus of adapting feature normalizations. We thank the reviewer for pointing out this work, and have included it in our revision.

---

> > ### Comment · AnonReviewer2 · 2018-11-19
> > **Response to your rebuttal**
> >
> > (a.) Frist, I do apologize for letting you think I was accusing you of plagiarism. This is a serious offense, and by no means I implied such a thing. While reviewing your paper and looking up the recent literature about Batch Normalization, I quickly came across the paper by Kalayeh & Shah, and I was surprised you didn’t mentioned it in your paper. I simply thought you had been scooped. I also apologize for not having taken a closer look (which I did now) at this paper.
> >
> > That said, I thank you for your detailed comment on the difference between both paper. As you mentioned, such comparison should figure in your literature review, since both methods are designed to provide multi-modality to BN. The key difference is indeed how it is implemented: They use an outside-of-the-loop GMM, while you use an attention mechanism. Your method is certainly easier to implement and use in modern deep learning frameworks than the GMM approach. A comparison with the GMM approach would still have been nice, or some histogram plots showing the means and variances of different modes.
> >
> > (b.) My point was that MN suffers even more than BN from the small size regime (note that this could also be a positive effect, as it could introduce stronger regularization). In Table 2, we can see that BN drops 3% error rate when going from 16 to 4 examples per mini-batch, where MN drops 4%. Also, this experiment is heavily multimodal in the first place (and thus one can expect BN to perform poorly, and this is the reason why I proposed (c.) for a more fair comparison). The gap in performances between MN and BN on CIFAR and ImageNet gets smaller and smaller, as the effective mini-batch size get smaller.
> >
> > Also by my comment that your paper "try to address BN’s weakness, which is an important direction in deep learning", I meant that your paper is going beyond uni-modal normalization, not that it is designed to solve the small size issue of BN.
> >
> > (c.) Sorry if I didn’t expressed myself clearly enough here. I was suggesting to use the information from which dataset D (MNIST, CIFAR, ...) one example comes from, and normalize it using the examples in the mini-batch that also come from dataset D. You would then obtain different statistics for different datasets. This would help to see how well your method compares against explicit separated normalization.
> >
> > (d.) I'm still interested to know if 1. you ran experiments on deeper networks (like the ResNet50) and 2. what is the validation sets you used through your experiments.
> >
> > I hope I let you enough time to answer again if you want to, and I will certainly increase the score of my review now that the difference between the two papers has been clearly established.

---

> > > ### Author Response · Authors · 2018-11-26
> > > **Re: Response to your rebuttal**
> > >
> > > (a.) Thanks for your reply and for the acknowledgement of significant differences between our paper and that of Kalayeh & Shah. Since there is no software available from the authors, and given the non-standard optimization technique and extensive hyperparameter tuning required to set it up, we leave a comparison as future work.
> > >
> > > (b.) Our focus is not to address all the weaknesses of batch normalization, but specifically to increase its robustness against multi-modality. Note however that we show that our model can be incorporated into group norm, which aims to address this issue. So in this sense we show that accounting for modality – as in mode group norm (MGN) – can increase robustness in a small batch size setting as well.
> > >
> > > Regarding (c.): Using an oracle to split batches via their original dataset is certainly possible, and results for this particular approach have previously been reported by Rebuffi et al. (2017). Since this approach does not make sense for the majority of our experiments (single task, where D=1), we excluded it from our evaluation. Using an oracle boosts the performance of LeNet by around 1-2%, but please note that this assumes both train and test time domain knowledge and cannot be used in single domain classification tasks.
> > >
> > > (d.) Our experiments involve tuning the learning rate schedule as well as the single additional hyperparameter of our method, K. For the former, we followed He et al. (2015) (p. 776 left of the CVPR version of their paper) in all experiments. For validating the latter, we randomly sampled 20% of the training set as validation and found K=2 to be a good compromise. After fixing K=2, we train our models on train+validation sets and report the result on test splits.
> > >
> > > As requested, we ran additional experiments on deeper networks, both on CIFAR10 and CIFAR100. For this, we implemented ResNet56 (which is more widely used for CIFAR tasks than ResNet50, see e.g. https://dawn.cs.stanford.edu/benchmark/CIFAR10/inference.html). Note that we used the exact same optimization setup as with ResNet20 in these experiments.
> > >
> > > On CIFAR10, ResNet56 with BN resulted in a test error of 6.87% (slightly better than the original result of 6.97% reported in He et al. (2015)). Replacing all normalization layers with MN achieves a test error of 6.47%, boosting the performance of BN by ±0.4%. Similarly, MN with ResNet56 obtains a test error of 28.69% versus 29.70% of BN, thus improves 1% over BN.

---

### Official Review · AnonReviewer1 · 2018-11-02
**Solid paper proposing a generalisation of Batch Normalisation**

**Rating:** 6
**Confidence:** 4

**Review:**

The paper proposes a generalisation of Batch Normalisation (BN) under the assumption that the statistics of the unit activations over the batches and over the spatial dimensions (in case of convolutional networks) is not unimodal. The main idea is to represent the unit activation statistics as a mixture of modes and to re-parametrise by using mode specific means and variances. The "posterior" mixture weights for a specific unit are estimated by gating functions with additional affine parameters (followed by softmax). A second, similar variant applies to Group Normalisation, where the statistics is taken over channel groups and spatial dimensions (but not over batches).

To demonstrate the approach experimentally, the authors first consider an "artificial" task by joining data from MNIST, Fashion MNIST, CIFAR10 and SVHN and training a classifier (LeNet) for the resulting 40 classes. The achieved error rate improvement is 26.9% -> 23.1%, when comparing with standard BN. In a second experiment the authors apply their method to "single" classification tasks like CIFAR10, CIFAR100 and ILSVRC12 and use large networks as e.g. VGG13 and ResNet20. The achieved improvements when comparing with standard BN are one average 1% or smaller.

The paper is well written and technically correct.

Further comments and questions to the authors:

- The relevance of the assumption and the resulting normalisation approach would need further justification. The proposed experiments seem to indicate that the node statistics in the single task case are "less multi-modal" as compared to the multi-task. Otherwise we would expect the comparable improvements by mode normalisation in both cases? On the other hand, it should be easy to verify the assumption of multi-modality experimentally, by collecting node statistics in the learned network (or at some specific epoch during learning ). It should be also possible to give some quantitative measure for it.

- Please explain the parametrisation of the gating units more precisely (paragraph after formula (3)). Is the affine mapping X -> R^k a general one? Assuming that X has dimension CxHxW, this would require a considerable amount of additional parameters and  thus increase the VC dimension of the network (even if its primary architecture is not changed). Would this require more training data then? I miss a discussion of this aspect.

- When comparing different numbers of modes (sec. 4.1, table 1), the size of the batch size was kept constant(?). The authors explain the reduction of effectiveness of higher mode numbers as a consequence of finite estimation (decreasing number of samples per mode). Would it not be reasonable to increase the batch size proportionally, such that the amount of samples per mode is kept constant?

---

> ### Author Response · Authors · 2018-11-26
> **Re: Solid paper proposing a generalisation of Batch Normalisation**
>
> We thank the reviewer for reading through our paper in detail. Three central concerns were raised: (a.) the modality should be quantified in some way, (b.) the parametrization needs to be explained in more detail, and (c.) experiments for constant N/K are missing.
>
> Regarding (a.), to the best of our knowledge no quantitative measure exists in the literature to describe the modality of a task. This is a very good question however, and to shed some light on it, we ran an additional analysis and evaluated the average standard deviation of intermediate features (per channel) in our VGG13 experiment. At test time, instead of transforming samples to a normal (with standard deviation 1), BN oversqueezes samples, with a mean deviation of ±0.5, considerably lower than the target of 1. MN yields a deviation of around 0.9, which lies much closer to the training target, so MN is better equipped to deal with modality at test time.
>
> (b.): Just as in standard BN, we compute estimators after average pooling over height and width of each image. As such, the affine transformation within the gating unit has its preimage in C. We realize now that the second paragraph on p. 5 was in need of some clarification, and have updated this in our revision. Many thanks for pointing this out to us!
>
> (c.): For this, please cross-reference the result for MN of Table 1 (where K=2, N=128, N/K=64) with Table 5 in the Appendix (K=4, N=256, N/K=64). While the gradient updates for the MN units (i.e. its estimators and the parameters of its transformations) receive equivalently informed gradients in both trials, the gradients for the convolutional layers differ, and in all likelihood the larger batch size of N=256 overdamps the gradient information for these layers. This overdamping issue is persistent even when doubling the number of training epochs.

---

### Official Review · AnonReviewer3 · 2018-11-07
**Normalization method that assumes multi-modal distributions**

**Rating:** 5
**Confidence:** 4

**Review:**

The authors proposed a normalization method that learns multi-modal distribution in the feature space. The number of modes $K$ is set as a hyper-parameter. Each sample $x_{n}$ is distributed (softly assigned) to modes by using a gating network. Each mode keeps its own running statistics.

1) In section 3.2, it is mentioned that the MN didn't need and use any regularizer to encourage sparsity in the gating network. Is MN motivated to assign each sample to multiple modes evenly or to a distinct single mode? It would be better to provide how the gating network outputs sparse assignment along with the qualitative analysis.

2) The footnote 3 showed that individual affine parameters doesn't improve the overall performance. How can this be interpreted? If the MN is assuming multi-modal distribution, it seems more reasonable to have individual affine parameters.

3) The overall results show that increasing the number of modes $K$ doesn't help that much. The multi-task experiments used 4 different datasets to encourage diversity, but K=2 showed the best results. Did you try to use K=1 where the gating network has a sigmoid activation?

---

> ### Author Response · Authors · 2018-11-19
> **Re: Normalization method that assumes multi-modal distributions**
>
> Many thanks for the review. Regarding 1): we consider MN to be a generalization of BN, and – see paragraph 4 on p. 5 – wanted to make sure the normalization unit can assume the standard form of BN, whenever that is optimal and yields the best performance. The obvious benefit of not regularizing this behavior is that MN becomes seamlessly insertable into any deep network. Regarding sparseness: note that (even at test time) assignments are usually quite pronounced, at roughly 0.95-0.99 on average.
>
> 2): Allowing individual affine parameters only improves test performance minimally (differences are in the regime of 0.05-0.2%). In all likelihood this is because normalizing features with multiple means and standard deviations already standardizes them sufficiently.
>
> 3): As shown in paragraph 2, p. 5, when K=1, MN reduces to standard BN. We also went ahead and implemented your suggestion to activate with a sigmoid. Unfortunately, the resulting performance was worse than that of vanilla BN.

---

### Public Comment · (anonymous) · 2018-10-01
**Details of experiments**

From table 1, it looks that increasing the number of K in MN also increases error rate. What value of K shall we use in practice?

---

> ### Author Response · Authors · 2018-10-02
> **Re: Details of experiments**
>
> Many thanks for your interest in our paper and your comment. Indeed, increasing the number of modes does not always increase performance, see also our third paragraph on p. 6.
>
> Intuitively, one would expect larger choices of K to always improve performance (at the expense of some computational cost). The fact that this isn’t the case connects to the same issue that also makes BN vulnerable to small batch sizes: for fixed N, increasing K results in less and less samples being assigned to a joint mode. Estimators are then computed from smaller partitions, in turn making them less accurate. Besides this, a second dynamic arguably comes into play in the hierarchicality of deep architectures. If the original network has L normalizations, then – compared to BN – we introduce L(K-1) additional normalizations in MN. So even in its simplest configuration, MN comes with L additional normalizations, which could be more than the network needs to account for the relevant modes in the distribution.
>
> In practice choosing K=2 gave us a significant performance boost in all our experiments (and therefore we recommend this value), going beyond that only resulted in benefits if the batch size was chosen to be sufficiently large, see the Appendix.

---

> > ### Public Comment · (anonymous) · 2018-10-02
> > **Re: Details of experiments**
> >
> > Thanks for reply. I still have a question. Are the examples normalized by the same mode in MN from the same category?

---

> > > ### Author Response · Authors · 2018-10-04
> > > **Re: Re: Details of experiments**
> > >
> > > Thank you for your continued interest. MN does not use any explicit label information, and (given the complexity of the datasets that we study here) is unable to uncover the underlying cluster structure, see penultimate paragraph on p. 5. Nonetheless, in our experiments we observe that MN does allocate samples into joint modes that have similar qualities, such as color or object size, c.f. Fig 2.

---

### Public Comment · ~Kun_Yuan1 · 2018-10-26
**About the Gating Network and Algorithm 1**

1. Since features in different layers represent differently, is there necessary to add a gating network alongside each normalization module? And what is the structure of your gating network?
2. Can you provide more details about Algorithm 1? Especially $y_{nk}$ and $x_n-\mu_k$，since different shape between (n,c,h,w) and (k,c) can not do subtraction directly.

---

> ### Author Response · Authors · 2018-10-29
> **Re: About the Gating Network and Algorithm 1**
>
> Hi, thanks for your interest and your questions. We parametrize the gating functions with an affine transformation followed by a softmax, see second paragraph on p. 5. Using an alternative in any subset of layers is certainly possible, this would need to be decided on a case-by-case basis though, as it depends on e.g. choice of architecture, or the task at hand.
>
> Regarding your second question, we apply the normalization to the full image, while estimators are computed after pooling over height and width, so we follow the exact same protocol as in batch norm.

---

### Meta-Review · Area_Chair1 · 2018-12-18
**Original generalization of batchnorm that yields small accuracy improvement.**

**Confidence:** 3
**Recommendation:** Accept (Poster)

**Metareview:**

The paper develops an original extension/generalization of standard batchnorm (and group norm) by employing a mixture-of-experts to separate incoming data into several modes and separately normalizing each mode. The paper is well written and technically correct, and the method yields consistent accuracy improvements over basic batchnorm on standard image classification tasks and models.
Reviewers and AC noted the following potential weaknesses: a) while large on artificially mixed data, improvements are relatively small on single standard datasets (<1% on CIFAR10 and CIFAR100)  b) the paper could better motivate why multi-modality is important e.g. by showing histograms of node activations c) the important interplay between number of modes and batch size should be more thoroughly discussed
d) the closely related approach of Kalayeh & Shah 2018 should be presented and contrasted with in more details in the paper. Also comparing to it in experiments would enrich the work.